# The Influence of Maximal Strength and Knee Angle on the Reliability of Peak Force in the Isometric Squat

**DOI:** 10.3390/sports9100140

**Published:** 2021-10-09

**Authors:** Arthur E. Lynch, Robert W. Davies, Philip M. Jakeman, Tim Locke, Joanna M. Allardyce, Brian P. Carson

**Affiliations:** 1Department of Physical Education and Sport Sciences, Faculty of Education and Health Sciences, University of Limerick, V94 T9PX Limerick, Ireland; Arthur@sigmanutrition.com (A.E.L.); robert.davies@ul.ie (R.W.D.); phil.jakeman@ul.ie (P.M.J.); Tim.locke123@hotmail.com (T.L.); joanna.allardyce@ul.ie (J.M.A.); 2Health Research Institute, University of Limerick, V94 T9PX Limerick, Ireland

**Keywords:** muscle strength, isometric contraction, measurement, reproducibility of results, strength testing

## Abstract

This study aimed to investigate the test-retest reliability of peak force in the isometric squat across the strength spectrum using coefficient of variation (CV) and intra-class correlation coefficient (ICC). On two separate days, 59 healthy men (mean (SD) age 23.0 (4.1) years; height 1.79 (0.7) m; body mass 84.0 (15.2) kg) performed three maximal effort isometric squats in two positions (at a 120° and a 90° knee angle). Acceptable reliability was observed at both the 120° (CV = 7.5 (6.7), ICC = 0.960 [0.933, 0.977]) and 90° positions (CV = 9.2 (8.8), ICC = 0.920 [0.865, 0.953]). There was no relationship between peak force in the isometric squat and the test-retest reliability at either the 120° (r = 0.052, *p* = 0.327) or 90° (r = 0.014, *p* = 0.613) positions. A subgroup of subjects (*n* = 17) also completed the isometric squat test at a 65° knee angle. Acceptable reliability was observed in this position (CV = 9.6 (9.3), ICC = 0.916 [0.766, 0.970]) and reliability was comparable to the 120° and 90° positions. Therefore, we deem isometric squat peak force output to be a valid and reliable measure across the strength spectrum and in different isometric squat positions.

## 1. Introduction

Assessments of maximal voluntary force (or strength) can provide valuable insight into one’s physical condition, differentiate strength characteristics between individuals, evaluate the efficacy of a training intervention, as well as inform exercise load prescription for future training programs [1]. However, there are several variables that can confound maximal force measurement (e.g., joint angle, angular velocity, use of the stretch-shortening cycle, and other skill-related components of the test). The use of isokinetic dynamometry can control for most of these confounding variables [1]. However, the external validity of these isolated, single-joint assessments in certain applications is low. Multi-articular assessments may be preferable to single-joint dynamometry, particularly for practitioners [2,3,4]. 

Traditional multi-articular measures of lower-body maximal strength such as the one-repetition maximum (1RM) back squat have much greater external validity compared to dynamometry-based assessments of strength [1]. However, such assessments may be unsuitable in certain circumstances due to both practical and methodological issues surrounding the conduct of 1RM back squat testing; in particular the control of the range of motion and the reliability of the test in certain populations [5,6]. The latter is highlighted by the susceptibility of 1RM to learning effects [6]. Furthermore, the reliability of 1RM testing tends to improve with training experience and/or baseline strength level [6]. Due to unfamiliarity with the exercise, it is unlikely that a true indication of muscle force output can be distinguished from the skill-related aspects of a 1RM test in untrained individuals. Moreover, 1RM testing can be quite demanding and poses a level of risk that could be deemed unacceptable [7]. For athletes, this may be a concern as repeated exposure to high loads may increase injury risk and/or residual fatigue [8,9].

The isometric squat (ISq) presents a credible alternative to the 1RM back squat as it is conducted at a fixed position, with acceptable levels of test-retest reliability for peak force (<10% CV, ≥0.8 ICC) [10,11]. The ISq test is less demanding than the 1RM, suggesting the ISq should provide good test-retest reliability across the strength spectrum. Herein, we operationally define the strength spectrum as covering the broad range of strength levels from untrained individuals up to and including highly strength-trained individuals. However, to the best of our knowledge, no previous study has investigated the reliability of peak force (PF) in the ISq across the strength spectrum. 

PF in the ISq is commonly assessed at knee angles of 120° and/or 90°, where full knee extension is 180° [12,13]. The 120° knee angle position (ISq_120_) closely replicates the strongest position of the back squat [14], whereas the 90° knee angle position (ISq_90_) approximates the ‘sticking point’ of the back squat (i.e., the point at which momentary muscular failure occurs during the lift) [11,15]. Although it produces lower peak forces compared to ISq_120_, the ISq_90_ correlates more strongly (r = 0.70–0.86) with dynamic 1RM back squat performance [10,11,16]. Whilst both test positions have been shown to produce reliable data for PF [10,11,16,17], previous ISq investigations have not explored the reliability of ISq positions that require knee angles of <90° [13]. Reasons for this may include difficulty attaining the position and/or lower PF reliability. Palmer et al. (2017) found isometric squat reliability reduced as the knee angle at which the test was performed lowered (i.e., CV at 150°, <120°, <90°), which may indicate that the reliability of the isometric squat at a knee angle of <90° may be inferior to that of higher knee angles [17]. Beyond this, it is difficult to speculate as, to the authors’ knowledge no data exist documenting the reliability of the isometric squat at a knee angle of <90°. However, it could be argued that in populations unfamiliar with deeper squatting positions (i.e., <90°), the ISq assessment may be safer compared to typical isoinertial assessments that demand the same range of motion (e.g., 1RM back squat). Additionally, assessment of strength in deeper squatting positions may be of relevance to athletes, particularly those involved in strength sports like (e.g., powerlifting and Olympic weightlifting) who regularly squat to knee angles of <90° [18]. 

The aim of the present study was twofold. Firstly, we aimed to investigate the relationship between ISq strength (as indicated by PF in the ISq), and test-retest reliability of PF in the ISq. It was a priori hypothesized that ISq reliability would be influenced by maximal ISq strength. Secondly, we sought to compare the reliability of PF in the ISq at a deeper squat position, which approximates the criteria for full depth in the back squat exercise (i.e., a 65° knee angle [ISq_65_]) [14,19,20,21]. We hypothesized that test-retest reliability of PF in the ISq would be correlated with baseline ISq strength and that both PF output in the ISq and test-retest reliability would be greater as the knee angle moved closer to full extension (i.e., ISq_120_ > ISq_90_ > ISq_65_).

## 2. Materials and Methods

### 2.1. Subjects

Prior to inclusion subjects were informed of the benefits and risks of participation before providing written informed consent. Eligibility criteria were: (i) men, (ii) 18 to 35 years of age, (iii) habitually active and in good general health with no current injuries, illness, or history of disease. In total, 59 men (mean (SD) age 23.5 (4.1) years, body mass 84.1 (15.2) kg, height 1.8 (0.7) m) were recruited from the local area and voluntarily took part in the study. Subjects’ training age ranged from 0 to 13 years; with relative 1RM back squat strength ranging from 0.7 to 2.7 kg∙kg^−1^ body mass. The sample consisted of 8 untrained subjects (no prior strength training experience whatsoever), 42 moderately strength-trained subjects (i.e., ≥6 months strength training experience, 1RM < 1.5 kg∙kg^−1^ body mass), and 9 highly strength-trained subjects (competitive powerlifters, ≥ 3 years strength training experience, 1RM ≥ 2.0 kg∙kg^−1^ body mass). A sub-group of 17 subjects (mean (SD) age 22.6 (2.6) years, body mass 81.9 (11.3) kg, height 1.8 (0.7) m) performed the ISq in a third position, at a 65° knee angle (ISq_65_). 

### 2.2. Instrumentation

ISq testing was conducted using a custom-made ISq rack (Odin Gym Equipment, Ireland) with a fixed barbell, adjustable in height, positioned above two force plates (AMTI, Watertown, MA, USA). The rack was bolted to the floor around the force plates (see Figure 1). Cortex motion analysis software (Rohnert Park, CA, USA) was used for the collection of PF data. For each contraction, ground reaction force data were sampled at 1 kHz and PF was defined as the highest value recorded from the force–time curve (excluding subject body weight). 

### 2.3. Procedures

Subjects reported to the lab at the same time (10:00 to 14:00) each test day to minimize diurnal variation. Eligibility and familiarization with the test procedures was undertaken 24 to 72 h prior to starting the study. Subjects were instructed to maintain their normal eating habits throughout the study. Dietary intake was recorded prior to the first test day and each subject was instructed to repeat this intake before the second test day. Subjects were instructed to refrain from any formal lower body exercise 48 h prior to testing and instructed to maintain their normal eating habits throughout the study. 

The subjects removed their shoes for the duration of the test procedures to control for any variation in footwear between subjects or between test sessions. Prior to commencing the warm-up, the rack heights that corresponded to the required ISq test positions were obtained. To do this, the subject assumed their preferred squatting position (i.e., stance width and foot position) on top of the force plates. The distance between the feet at the anterior (i.e., distance between the two first distal phalanges) and posterior extremities (i.e., distance between the most posterior and medial aspect of the left and right calcaneus bones) was measured, recorded, and marked with tape. For repeat trials, tape was re-laid using the measurements recorded in the first testing session, allowing for consistency across trials. Whilst maintaining this same squatting stance, each subject was then instructed to descend until the desired knee angle (i.e., 120°, 90°, or 65°) was obtained. The knee angle was measured using a plastic goniometer (Fabrication Enterprises, White Plains, NY, USA) which was placed on the lateral condyle of the femur, the fixed end was aligned with the greater trochanter of the femur, and the moving arm aligned with the lateral malleolus of the ankle. Additionally, angles at the hip and ankle joints were recorded to ensure within-subject consistency across trials.

The testing order for the two positions (ISq_120_ and ISq_90_) was randomized for each subject on the first test day and this same order was repeated on the second test day. Subjects completed three warm-up ISq contractions in the first test position prior to measurement (50%, 70%, and 90% of perceived maximal effort) which were maintained for 3 s with a 1 min rest provided between each sub-maximal contraction. Immediately following the sub-maximal contractions, a 3 min rest period was provided before the first maximal effort contraction and then again for each subsequent maximal effort contraction [10]. Subjects were given standard verbal encouragement from the investigators, who instructed subjects to “*push as hard and fast as possible into the bar*” and to maintain peak force output for the duration of the 4 s contraction. Three maximal effort ISq contractions were performed in each position. Vertical ground reaction force data were sampled at 1 kHz and excluded if any countermovement was evident. PF was determined as the highest value attained out of the three attempts on each test day. All subjects performed three maximal effort ISq contractions at ISq_120_ and ISq_90_ on two separate test days at least 72 h apart (range: 72–168 h). A minimum of 72 h rest was provided between test days, which has been evidenced to allow full recovery from strenuous lower body dominant resistance exercise [22]. A sub-group of 17 subjects also performed an additional three maximal effort ISq contractions at ISq_65_ on each test day.

### 2.4. Statistical Analyses

Test-retest (i.e., day 1 vs. day 2) reliability scores were calculated for PF for all subjects, analyzed as one group (*n* = 59). Normality and homogeneity of variance were assessed prior to analysis (Shapiro Wilk and Levene’s test respectively). A two-way random model with absolute agreement was used to calculate the intra-class correlation coefficient (ICC) [23]. An ICC over 0.9 was defined as highly reliable, between 0.8 and 0.9 as moderately reliable, and below 0.8 as not reliable [23,24,25]. 

The between-day coefficient of variation (CV) was calculated. A CV of < 10% is accepted as reliable [6]. Paired samples *t*-tests were used to assess differences between days (day 1 vs. day 2) for PF and between positions (ISq_120_ vs. ISq_90_) for both PF and CV, with the alpha level set at *p* < 0.05. To assess the relationship between maximal ISq strength and the reliability of PF measurement, a correlation analysis (Pearson’s correlation coefficient) was conducted in Microsoft Excel by plotting subjects’ PF scores at ISq_120_ and ISq_90_ against their respective CV scores in each position. 

To assess the influence of knee angle on PF, a one-way ANOVA was performed to compare differences in PF between positions (ISq_120_ vs. ISq_90_ vs. ISq_65_), with the alpha level set at *p* < 0.05. Normality and homogeneity of variance were assessed prior to this analysis. This same analysis was also performed with subjects divided into their respective sub-groups based on training experience (i.e., untrained vs. moderately trained vs. highly trained. To assess the influence of knee angle on ISq reliability, a one-way ANOVA was performed to compare differences in CV between positions. Where significant differences were detected, paired sample *t*-tests were used to identify differences between knee angles. Effect size (*d*) was calculated for differences in PF between positions by dividing the position difference by the pooled standard deviation [25]. The effect sizes were classified as small (0.2), medium (0.5), or large (0.8) [26]. The between-day difference with upper and lower (95%) levels of agreement (LOA) was used to assess systematic bias. Typical error (TE) was calculated for PF by dividing the standard deviation of the within- subject differences in PF between the two test days (i.e., the peak of day 1 and the peak of day 2) by √2 [27]. TE provides an indication of the change score required between trials to be considered “real” (i.e., outside the normal range of error associated with the test) [27]. As such, the TE could be considered an index of the sensitivity of the measure to detect change over time and provides real value for practitioners who may consider using the ISq to measure performance changes in response to a training program [11,27].

## 3. Results

### 3.1. Relationship between Isometric Squat Strength and Measurement Reliability

Reliability data for all subjects (*n* = 59) are presented in Table 1. PF was significantly greater at ISq_120_ than at ISq_90_ (*p* < 0.001). All variables achieved a high/acceptable level of reliability (ICC > 0.9; CV < 10%) at both ISq_120_ and ISq_90_ (Table 1). Furthermore, when the analysis was performed on the subject sub-groups (untrained vs. moderately trained vs. highly trained), no significant differences in measurement reliability were observed in either position (*p* ≥ 0.146). Correlation analyses of the test-retest reliability revealed no relationship between maximal ISq strength and ISq reliability at either ISq_120_ (r = 0.052, *p* = 0.327) or ISq_90_ (r = 0.014, *p* = 0.613) as shown in Figure 2.

### 3.2. Relationship between Isometric Squat Position and Measurement Reliability

The test-retest reliability of PF measurement at ISq_120_, ISq_90_ and ISq_65_ in the subject sub-group (*n* = 17) is presented in Table 2. PF was significantly greater at ISq_120_ compared to ISq_90_ (*p* < 0.001, *d* = 0.8) and ISq_65_ (*p* < 0.001, *d* = 1.0). There was no significant difference in PF between the ISq_90_ and ISq_65_ positions (*p* = 0.052, *d* = 0.3). Results of the one-way ANOVA revealed that test-retest reliability was not different between positions (ISq_120_ CV = 7.3%; ISq_90_ CV = 8.9%; ISq_65_ CV = 9.6%, *p* = 0.6) and an acceptable level of reliability was observed in all positions based on ICC values (ISq_120_ ICC = 0.969; ISq_90_ ICC = 0.892; ISq_65_ ICC = 0.916) Mean between-day difference (day 2–day 1) was determined via Bland–Altman analysis for PF in each position (Figure 3). No systematic bias was observed between test days for any variable or position (Figure 3). Finally, the TE values were determined for PF in each position (ISq_120_ = 118 N (10.5%); ISq_90_ = 117 N (14.2%); ISq_65_ = 83 N (11.0%), see Table 2).

## 4. Discussion

Training status and by association strength has previously been shown to influence the reliability of strength measures such as 1RM squat and leg press, with greater reliability typically observed with increased strength and/or strength training experience [6,28,29]. Though previous literature has documented the reliability of a variety of ISq tests across populations with different levels of strength, no previous study has investigated the relationship between maximal ISq strength and the reliability of the measure [10,11,16,17]. Therefore, the initial aim of this study was to investigate the test-retest reliability of the ISq across the strength spectrum. Here we report acceptable levels of reliability for PF in a sample of 59 subjects (Table 1) based on previously established reliability cut-offs (i.e., >0.8 ICC and <10% CV [6,24,30]. Correlation analyses revealed no relationship between maximal ISq strength (as indicated by PF) and ISq reliability at ISq_120_ and ISq_90_ (Figure 2). It is worth noting however that based on Figure 2, all outliers (i.e., CV ≥ 15%) seem to occur amongst individuals with lower peak force values.

Though no previous studies have directly investigated the relationship between maximal ISq strength and the reliability of ISq measurement, ISq reliability studies across different populations with distinct levels of strength are available to place the findings of this study in context. Tillin et al. (2013) is the only available study that measured ISq force variables in two distinct subject sub-groups of differing strength levels (18 elite male athletes and 8 untrained male subjects, with the ISq conducted at a mean knee angle of 118°), however, reliability data were only reported for the untrained subjects (CV = 4%, ICC = 0.960) [31]. In a group of recreationally trained male subjects (≥ 1 year training experience), similar ICC values (0.97 and 0.99 at ISq_120_ and ISq_90_, respectively) were reported by Bazyler et al. (2015) [10]. Additionally, Drake et al. (2018) reported comparable CV and ICC values of 6% and 0.856 at ISq_90_ in a group of strength-trained males (mean (SD) training experience 4.1 (1.8) years) [11]. Moreover, Blazevich et al. (2002) reported an ICC of 0.97 in the ISq_90_ in a group of athletic men [16]. This suggests that training status, and by association maximal strength, may not influence ISq reliability. However, Palmer et al. (2017) reported CV and ICC values of 11.2% and 0.839 respectively at ISq_120_ as well as 12% and 0.885 at ISq_90_, respectively, in a group of resistance-trained females, which differs quite considerably from the rest of the available literature [17]. The reason for this is not readily apparent, though it may be related to the specific isometric squat apparatus used in this investigation [17]. The data reported here display similar levels of reliability and supports the observation that there is no relationship between maximal ISq strength and test-retest reliability of the ISq. This observation is consistent across different positions (e.g., ISq_120_, ISq_90_, ISq_65_), as reported here and elsewhere [10,11,16,17,31].

The second objective of this study was to determine the reliability of PF measurement in the ISq_65_, compared to the ISq_120_ and ISq_90_, in a sub-sample of eligible subjects. To the authors’ knowledge, this is the first study to investigate the use of an ISq position at a 65° knee angle, which is somewhat surprising given it closely approximates the bottom position of the back squat (Hales et al., 2009; Swinton et al., 2012). All positions including ISq_65_ displayed acceptable reliability based on ICC values (>0.8) [23,24,25]. We noted that, whilst it was reliable and not significantly different from ISq_120_ and ISq_90_ positions, the CV was highest at ISq_65_ (Table 2). Furthermore, based on researcher observation and verbal feedback from the subjects, the ISq_65_ position was unfamiliar and as a result, subjects reported that it was more difficult to establish. Given the increased mobility demands of the ISq_65_, some subjects had difficulty attaining the required positioning, which may have affected the reliability of the force output in this position. We speculate that in contrast with the ISq_65_ position, the ISq_120_ and ISq_90_ positions require minimal skill and mobility, allowing subjects to focus their efforts on exerting maximal force into the bar. That said, it is also plausible that a greater number of familiarization sessions may have improved the reliability of the force output in the ISq_65_ position [11] and, therefore, may be worth consideration for future research. Drake et al. (2018) observed improvements in PF in the ISq_90_ past three test days, indicating that more sessions may improve the reliability of the measure at each position [11]. 

TE values for each ISq position were calculated to indicate the change scores required over time to be considered real (Table 2). These values allow researchers and practitioners alike to determine what is the smallest change required over time to be considered real if they are working with subjects from the same population from which our sample was drawn. This is a valuable consideration for those using the ISq to monitor performance changes in response to training interventions. Based on our data, change scores of 10.5%, 14.2%, and 11.0% are required to be considered real changes at ISq_120_, Isq_90_, and ISq_65_ respectively. 

Overall, we demonstrate acceptable levels of reliability in an ISq test conducted at knee angles of 120° and 90°, providing evidence that the ISq can be used to reliably measure PF. Additionally, we observed no relationship between the maximal strength of an individual and the reliability of ISq measurement. This is an important finding for sport science researchers and practitioners as it adds merit to the applicability of ISq measurement across athletes of different strength levels. For clinical research, the ISq may also have application as a measure of maximal strength in studies of sedentary/untrained subjects. For practitioners, the ISq offers a potential alternative to typical 1RM testing, which can be difficult to standardize and carries an inherent level of risk that may not be deemed acceptable. In addition, the data indicate that the ISq is a reliable test of maximal strength when conducted at a knee angle of 65°. However, it is important to note that this position is more difficult to attain and researchers may be best served by conducting the ISq at a 90° or a 120° knee angle, and this is perhaps the reason this data are more commonly reported in the literature.

## Figures and Tables

**Figure 1 sports-09-00140-f001:**
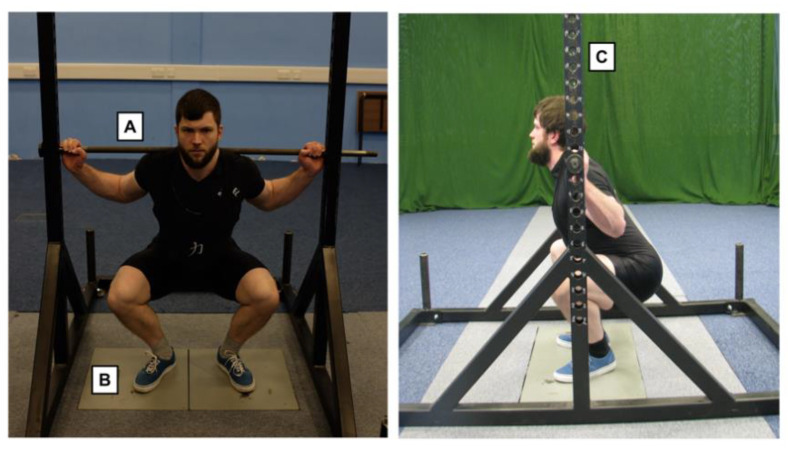
Isometric squat rack apparatus and subject setup: (**left**) = front view, (**right**) = side view. A = fixed barbell, B = AMTI force plates, C = adjustable bar heights (2.5 cm between rack heights). In this example, the isometric squat was performed with knee and hip angles of 65° and 115°, respectively.

**Figure 2 sports-09-00140-f002:**
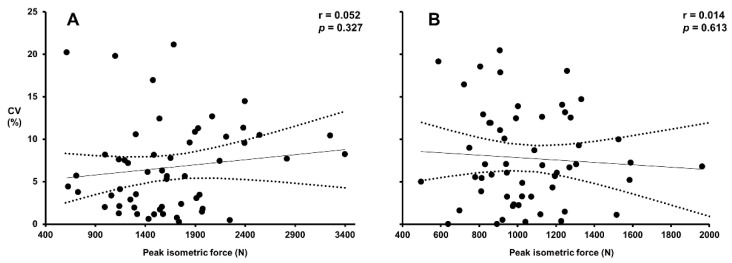
Correlation analyses of maximal ISq strength (PF) and reliability via CV in the isometric squat: (**A**) represents the reliability of PF (highest values of day 1 and day 2) at ISq_120_, plotted against the corresponding CV values. (**B**) Represents the reliability of PF (highest values of day 1 and day 2) at ISq_90_, plotted against the corresponding CV values. Dashed lines indicate the 95% confidence intervals.

**Figure 3 sports-09-00140-f003:**
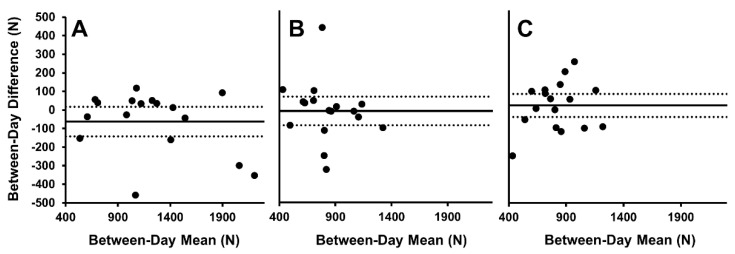
Bland–Altman plots for between-day peak isometric force. (**A**) PF at ISq_120_, (**B**) PF at ISq_90_, and (**C**) PF at ISq_65_. Solid line represents the mean difference; dashed lines represent 95% limits of agreement.

**Table 1 sports-09-00140-t001:** Reliability of peak isometric force at ISq_120_ and ISq_90_ (*n* = 59).

Peak Force (N)				
	Day 1 [Mean (SD)]	Day 2 [Mean (SD)]	Difference (Day 2–Day 1) Mean [LLOA, ULOA]	
ISq_120_	1579 (536)	1536 (567)	−44 [−100, 12]	
ISq_90_	1035 (298) ^a^	958 (275) ^a^	−73 [−114, −32]	
**Test-Retest Reliability**				
	**CV [Mean (SD)]**	**ICC [95% CI]**	**TE (N)**	**TE (%)**
ISq_120_	7.5 (6.7)	0.960 [0.933, 0.977]	140.2	9.5
ISq_90_	9.2 (8.8)	0.920 [0.865, 0.953]	82.6	8.8

LLOA = Lower limits of agreement, ULOA = Upper limits of agreement, CV = coefficient of variation, ICC = intraclass correlation coefficient, TE = Typical error, 95% CI = 95% confidence interval, a = significantly different peak isometric force from the ISq_120_ position (*p* < 0.001).

**Table 2 sports-09-00140-t002:** Reliability of peak isometric force across isometric squat positions (*n* = 17).

Peak Force (N)				
	Day 1 [Mean (SD)]	Day 2 [Mean (SD)]	Difference (Day 2–Day 1) Mean [LLOA, ULOA]	
ISq_120_	1259 (525) _a,b_	1197 (467) _a,b_	−63 [−142, 17]	
ISq_90_	919 (287) _c,b_	848 (288) _c_	−50 [−120, 21]	
ISq_65_	820 (205) _c,a_	826 (233) _c_	6 [−53, 66]	
**Test-Retest Reliability**				
	**CV [Mean (SD)]**	**ICC [95% CI]**	**TE (N)**	**TE (%)**
ISq_120_	7.3 (4.3)	0.969 [0.914, 0.989]	118	10.5
ISq_90_	8.9 (6.7)	0.892 [0.702, 0.961]	117	14.2
ISq_65_	9.6 (9.3)	0.916 [0.766, 0.970]	83	11.0

LLOA = Lower limits of agreement, ULOA = Upper limits of agreement, CV = Coefficient of variation, ICC = Intraclass correlation coefficient, 95% CI = 95% confidence interval, TE = Typical error, a = significantly different (*p* < 0.05) from ISq_90_, b = significantly different from ISq_65_, c = significantly different from ISq_120_.

## Data Availability

The data presented in this study are available on request from the corresponding author.

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
