# Peer review of "The Influence of Maximal Strength and Knee Angle on the Reliability of Peak Force in the Isometric Squat"

_sports, 2021, doi:10.3390/sports9100140_

Round 1

Reviewer 1 Report

Summary:

A repeated measures design was used to observe the relationship between peak forces and knee joint angle while performing maximal effort isometric squats. Isometric squats were performed in two primary positions, and another group performed this task at a third knee joint angle, on two separate days.

Comments:

The authors have indicated that isometric squats are reliable in determining peak force output. There are a number of items that the authors must address. First, there was the assumption that the experience of the participants was not a factor that would influence the results. This, however, was not tested – and the authors also reported that CV values were higher in the untrained individuals, or those with the lower PF values. Second, would it be better to test forces generated in isometric squats to those while performing a squat exercise?

Introduction:

  • The authors are speculating about PF in knee angles less than 90 degrees, but this did not appear to be a large objective of the study. If the reliability of the PF measures is the focus, then the authors need to maintain this point in their study. Further it is unclear that the 65 degree angle was a scientifically sound choice for the sub-sample as the muscles rely mostly on the stiffness of the tendons and other connective tissues at this squat depth

Methods:

  • Why not compare the three different groups of individuals? If there was no difference indicated, then the data can be collapsed
  • The position of the knee joint and foot positions were controlled, but it is not clear if the position of the trunk was controlled. This would influence the length of the hip extensors

Discussion:

  • Lines 268-269: it is not clear what the authors mean by this comment. The CV of females was greater in one study, but the authors cannot state that this was due to the sex of the participants – what were the parameters of the study design?

Reviewer 2 Report

Review

The paper is interesting and covers a lot of useful and practical information. It needs however many clarifications on facts mentioned in tekst. The title is not exactly combined with the analysis described in the paper. I think the title is not accurate, there is a mild exaggeration. Instead of ,,reliability of peak force in the isometric squat’’ Your paper shows effect of chosen angle positions on PF and CV, because that was investigated.

The critical part of review on your own research is also missing. The method part should be written in more detailed way. It is full of information and difficult to follow. Probably definitions and test should be written in table. It seems clear that many limits and data are given as citation, but own performed results are not enough described.

Especially statistical analysis part is full of unclear information. It is not clear what kind of data are tested by paired t-tests to assess differences between days and positions (L167). CV?

The analysis of correlations between PF and ISq120 and ISq90 was done by plotting?

Why coefficients of correlations were not used – for example simple Pearson correlation? Such info allows recognition of power and significance of correlations.

There are many shortcuts – L 172 – to assess the influence of ISq angle on …? The method part should be as clear as possible for every reader, without shortcuts. The procedure parts describe in detail all tests, however in the end it should be underlined what data/measurments were received and how were they treated in the statictical part, in detail. There are also some shortcuts – L 175. Effect size was calculated? Please give details/procedures.

Other remarks below:

L 250 – all outliers are weaker individuals? Why do you state so? Please give clarifications.

L 264 – 271 I am not sure if your discussion is not going too far at this point. What about the mass of the investigated men? It was not discussed at all and the variability of the body mass is big. Have you checked the influence of the body mass/composition of your tests results? Usually the trained subjects change the variability of their results (also biomechanical) with the training. The coefficient of variation can be a good indicator of training status and health. Repeatable results on wanted level are the sign of favourable performance. Your investigations were conducted on different groups of experience? I do not see where was it taken into account in your method part. The analysis of variance with all effects possible to identify should be conducted in my opinion.

Possible influence of the period between tests is not discussed.

Special attention should be put in the discussion in comparison between knee position 65 and the others (120,90) as the investigated group was not the same. Please make the note on it in discussion, the statistic power is weaker in smaller groups.

It is difficult to discuss the paper further without detailed knowledge on statistics used. In the presented version the statistical analysis seem to be not enough. The effect of training experience (groups - 8 beginners, 42 moderately trained and 9 highly trained subjects are not taken into account). The group of 17 subjects is not described in performance. The method of correlations analysis not clear, and seems not enough. 

Round 2

Reviewer 2 Report

Review  Sport 1327616 v2

The paper was corrected meaningfully, however most important methodological points are still omitted.

Please correct your data on experience of the athletes. It can be done in different ways:

  1. analysis of variance with all factors - knee angle, maximal strength, day, experience (in classes of beginners, middle experience and highly strength / or check the regression on 1RM in the model?)

Or at least

  1. analysis in experience subgroups (your present analysis can be kept for comparison, however the number of athletes is not the same)

The   Authors should also think again about the statistical methods  – if the CV for 65°knee angle is mean 9.6 with sd 9.3, so the distribution does not see normal. The statistics they used should be used for normal data distribution.  What about data distribution? Perhaps non-parametrical tests should be used?

The kind of correlation received by plotting (what statistical program? procedure?) is not named. I can be simply Pearson correlation, but it has to be named in the paper. Please check your program guide to know the kind of correlations.

The answer to reviewer about period between tests with citation should be included in the manuscript.

Detailed remarks:

Discussion:

L 298-299 your data can be biased by the experience, so that was not investigated.

L 304-305 the correlation inform about relation, not the influence; so that is incorrect

L 310-314 – that is not checked in the study, it is practitioner’s information – it may be very useful, however in the discussion. The last part of your paper should be based on studied, statistically proved  analysis.

Round 3

Reviewer 2 Report

Please include all details on normal distribiution testing in to the text of the paper (not only in the review).